# Impact of the Interaction of Hepatitis B Virus with Mitochondria and Associated Proteins

**DOI:** 10.3390/v12020175

**Published:** 2020-02-04

**Authors:** Md. Golzar Hossain, Sharmin Akter, Eriko Ohsaki, Keiji Ueda

**Affiliations:** 1Division of Virology, Department of Microbiology and Immunology, Graduate School of Medicine, Osaka University, Osaka 565-0871, Japan; eohsaki@virus.med.osaka-u.ac.jp; 2Department of Microbiology and Hygiene, Bangladesh Agricultural University, Mymensingh 2202, Bangladesh; 3Department of Physiology, Bangladesh Agricultural University, Mymensingh 2202, Bangladesh; sharmin.akter@bau.edu.bd

**Keywords:** Hepatitis B virus (HBV), mitochondria, proteins/signaling, interaction

## Abstract

Around 350 million people are living with hepatitis B virus (HBV), which can lead to death due to liver cirrhosis and hepatocellular carcinoma (HCC). Various antiviral drugs/nucleot(s)ide analogues are currently used to reduce or arrest the replication of this virus. However, many studies have reported that nucleot(s)ide analogue-resistant HBV is circulating. Cellular signaling pathways could be one of the targets against the viral replication. Several studies reported that viral proteins interacted with mitochondrial proteins and localized in the mitochondria, the powerhouse of the cell. And a recent study showed that mitochondrial turnover induced by thyroid hormones protected hepatocytes from hepatocarcinogenesis mediated by HBV. Strong downregulation of numerous cellular signaling pathways has also been reported to be accompanied by profound mitochondrial alteration, as confirmed by transcriptome profiling of HBV-specific CD8 T cells from chronic and acute HBV patients. In this review, we summarize the ongoing research into mitochondrial proteins and/or signaling involved with HBV proteins, which will continue to provide insight into the relationship between mitochondria and HBV and ultimately lead to advances in viral pathobiology and mitochondria-targeted antiviral therapy.

## 1. Introduction

Mitochondria are double-membraned organelles of eukaryotic cells. The basic structure of the mitochondrion is composed of an outer mitochondrial membrane (OMM), inner mitochondrial membrane (IMM), intermembrane space, cristae, and matrix. There are 1000–2000 different proteins in the mitochondria [1]. The most important functions of mitochondria are to generate energy, regulate cell metabolism, and transmit calcium signaling to other organelles in the cells. Mitochondria also play crucial roles in cell apoptosis [2]. The OMM can associate with the membrane of the endoplasmic reticulum (ER), resulting in a structure called the mitochondria-associated ER-membrane (MAM). This communication is important for calcium signaling between the OMM and ER [3]. The number of mitochondria varies dependent on the cell types. Liver cells contain 500–4000 mitochondria per cell [4]. One study suggested that mitochondria play important roles in the innate immune system against viral infection through the mitochondrial antiviral signaling protein (MAVS) [5]. 

Mitochondria can change their shape and location, and can undergo fusion, fission, and mitophagy in response to cellular stresses in order to maintain homeostasis [6]. These mitochondrial dynamics might be dependent on different pathological conditions, such as cancers and viral infection [6,7,8], and many viruses—including hepatitis B virus (HBV), hepatitis C virus (HCV), pseudorabies virus, human cytomegalovirus (HCMV), Epstein–Barr virus (EBV), influenza A virus, measles virus, Newcastle disease virus (NDV), and SARS corona virus [6]—may interfere with the mitochondrial dynamics. These viruses have been found to interact with many mitochondrial proteins and disrupt mitochondrial dynamics, resulting in cellular apoptosis, intracellular calcium signaling, and innate immune signaling [6].

HBV is a partially double-stranded DNA virus belonging to the family of hepadnaviridae, and causes an acute and chronic liver disease known as hepatitis B (HB) [9,10]. HBV contains a 3.2 kb genome surrounded by an icosahedral capsid and an envelope. Its genome encodes four overlapping open reading frames (ORFs) known as polymerase (pol; P), surface or envelopes (S), core (C), and X protein (X) (Figure 1) [11,12,13,14]. HBV enters hepatocytes through sodium taurocholate cotransporting polypeptide (NTCP) as a receptor to bind with the preS1 region in the envelope protein, and then the uncoated nucleocapsid is transported to the nucleus, where the relaxed circular DNA (rcDNA) genome is converted to covalently closed circular DNA (cccDNA) [15,16,17]. The cccDNA is competent for transcription of 3.5 kb pregenomic RNA (pgRNA), and for transcription of several subgenomic RNAs such as 2.4 kb preS-S mRNA, 2.1 kb S mRNA, and 0.7 kb X mRNA. The pgRNA is encapsidated together with pol and then reverse- transcribed into negative strand DNA, resulting in the formation of rcDNA. This newly formed nucleocapsid re-enters the nucleus as a result of intracellular cycling [17,18]. 

HBV encounters many subcellular organelles and host factors/proteins during the viral life cycle, including those involved in entry, nuclear transport of capsids, DNA replication, assembly, and egress [19]. HBV also targets mitochondria and disrupts mitochondrial dynamics, and several viral proteins localize at the mitochondria and interact with numerous mitochondrial proteins [7,20,21,22,23].

Recently, host factors involved in mitochondrial turnover have also been targeted for the HBV treatment [24]. Therefore, it would be useful to elucidate further details and basic information about the relationship between HBV and mitochondria. In this review, we summarize and discuss the recent literature on the interaction of mitochondrial factors and HBV gene products. 

## 2. Methods

We comprehensively reviewed published articles related to HBV and mitochondria, and associated proteins. The key words we used to search the literature were “hepatitis B virus and mitochondria”, “HBx and mitochondria”, “HBV polymerase and mitochondria”, “HBsAg and mitochondria”, “HBV core and mitochondria”, “preS1 and mitochondria”, and so on. The related articles cited in the searched articles were also reviewed.

## 3. HBV and Mitochondria

It was reported that leakage of endoplasmic reticulum (ER)-calcium stores was caused by ER- stress on HBV infection and adjacent depolarized and/or dysfunctional mitochondria, leading to ROS generation [25]. It was also reported that HBV might alter mitochondrial dynamics, leading to mitochondrial injury of hepatocytes, and subsequently liver disease onset [7,25]. 

### 3.1. HBx

HBx is a multifunctional protein encoded by ORF X and enhances HBV replication [26]. It activates various cellular transcription factors and plays roles in cell cycle regulation, calcium signaling, DNA repair, apoptosis regulation, ROS regulation, etc. [26,27,28]. Many studies reported that HBx is localized in mitochondria, either in the OMM, IMM, or matrix [29,30,31,32,33]. It has been demonstrated that the C-terminal transactivation domain of HBx [34] and the amino acids 54 to 70 of HBx [30] are involved in its mitochondrial localization in Huh7 and WRL68 cells transiently transfected with tagged based HBx, as shown by immunofluorescence analyses (IFA). This mitochondrial localization of HBx has been further confirmed by mitochondrial purification assay from HepG2 2.2.15 cells, a cell line persistently expressing a dimer molecule of HBV DNA under the control of native promoter [30]. Clippinger and Bouchard demonstrated that HBx was partially localized at the OMM in primary rat hepatocytes and HepG2 cells transfected with HBx and it regulates mitochondrial membrane potential (Δψm) and activated NF-κB [32]. However, it is still unclear how mitochondria-associated HBx regulates HBV replication. 

HBx induces production of reactive oxygen species (ROS), including mitochondrial ROS (mROS). Increased mROS damages mitochondrial DNA (mtDNA) that might play a role in hepatocellular carcinoma (HCC), and HBx-mediated ROS generation activates transcription factors such as Foxo-4 in Chang cells stably expressing HBx (Chang-HBx) and in primary hepatic tissues from HBx-transgenic mice; STAT-3, as well as NF-κB in HBx-transfected HepG2 cells [27,35,36,37]. Alteration of mtDNA has been reported in chronic HBV patients, and could be associated with HCC [38,39,40,41]. Chen et al. found significantly higher mtDNA in peripheral blood leukocyte (PBL) of chronic hepatitis B patients compared with a healthy control [39]. Moreover, higher mtDNA have been quantified in serum samples of HBV patients but revealed increased risk of HCC development with lower mtDNA level [38]. Zhao et al. also reported that lower mtDNA in PBL lead to increased risk of HCC [40]. The mtDNA copy number in HCC tissue samples correlated with large tumor size and liver cirrhosis [42]. A recent clinical study detected considerably higher mitochondrial superoxide in the cells of chronically infected patients and found extensively altered mitochondria [43]. Increased intrahepatic lipid peroxidase has been also reported in HBx-transgenic mice [44]. HBx directly binds with Raf-1 and stimulates its translocation to mitochondria in Huh7 cells transfected with pCMV4X, suggesting that mitochondrial Raf-1 should protect cells from apoptotic stress [45]. Several studies showed that HBx colocalized with COXIII, an inner mitochondrial membrane protein, and upregulated its expression in HepG2 cells, and that HBx elevated ROS and altered mitochondrial biogenesis and morphology [46,47,48,49]. Endogenous COXIII colocalized with HBx in HepG2 cells generated by lentivirus transduction, which leads to the upregulation of COX-2 expression, thereby promoting cell growth [46]. HL-7702 cells stably expressing HBx generated by lentivirus transduction (HL-7702-HBx) led to swollen mitochondria, as shown by transmission electron microscopy [49]. Yoo et al. analyzed the subcellular distribution of HBx-Flag in Huh7 cells by confocal microscopy at different time points after transfection [50]. At 24 h after transfection, the HBx was found to be localized mainly in the nucleus and partly in the cytoplasm. However, HBx accumulated in the cytoplasm and in mitochondria at 36 h, and more than 50% of the HBx localized into mitochondria as dot-like aggregates at 48 h after transfection. A mitochondrial E3 ubiquitin ligase, MARCH5 interacted and colocalized with HBx in mitochondria and promoted the degradation of HBx aggregates by polyubiquitination in Huh7 cells co-transfected with Myc-MARCH5 and mitochondria-targeted HBx (HBx-Mito-Flag) [50]. MARCH5 localized on OMM and regulated mitochondrial dynamics by ubiquitinating several mitochondrial proteins such as Drp1, Fis1, and Mfn1. MARCH5 along with Mfn1 maintain mitochondrial homeostasis and cell survival [51,52,53,54]. Moreover, MARCH5 decreased the HBx-induced NF-κB and COX-2 activity which may play important roles in carcinogenesis [50,55,56]. The authors further demonstrated that MARCH5 mRNA and protein expression in either HCC or HBV-mediated HCC liver tissue specimens of clinical cases were significantly downregulated in a later stage (Stage IV) of cancers with high expression of HBx [50]. In addition, AIM2 (absent in melanoma2) protein could be involved in cell proliferation and tumorigenic reversion and Aim2 deficient mice are more susceptible to the development of colonic tumor [57,58]. HBx reduced the expression of AIM2 which leads to HCC metastasis through the activation of EMT (epithelial-mesenchymal transition) by increasing expression of mesenchymal markers, vimentin, and N-cadherin and decreasing expression of E-cadherin, an epithelial marker in AIM2-overexpressed Bel-7402 and SMMC-7721 cells [59]. These results strongly correlated with the clinical cases as AIM2 expression has been found at significantly low level in tissues of HCC patients [59].

Many reports have suggested that HBx induces apoptosis. Takada et al. reported that HBx strongly interacted with p53 in the aggregated mitochondrial structure in tranfected Huh7 cells, probably at the OMM, and led to cell death [29]. Other reports suggested that HBx interacted with the human voltage-dependent ion channel (hVDAC3), which is an outer mitochondrial protein, and decreased Δψm of transfected HepG2 cells and cultured primary rat hepatocytes, thereby leading to cell death [21,22]. HBx increases cytosolic calcium by regulating the hVDAC component, mitochondrial permeability transition pore (MPTP) of HepG2 cells transfected with an HBV replication competent plasmid payw1.2 under control of endogenous promoter [60,61]. Moreover, HBx interacts with endogenous Bax in HepG2 cells stably expressing HBx, which acts as a pro- apoptotic regulator, and enhances the translocation of Bax to mitochondria, the release of cytochrome c, and the induction of apoptosis [62,63]. Bax itself also interacts with hVDAC and causes a reduction of Δψm and release of cytochrome c [31,33]. Cardiolipin (CL), a mitochondrial lipid, is predominantly located at IMM and plays important roles in mitochondrial functions, including apoptosis and mitophagy [64]. You et al. demonstrated that HBx bound with CL and increased membrane permeabilization [65]. However, Lee et al. reported that HBx did not activate apoptotic signaling in transfected HepG2 cells though it increases mROS [44], and none of these above studies showed the involvement of mitochondria. 

HBx upregulates the expression of mitochondrial serine/threonine-protein kinase (PINK1) [7]. PINK1 is localized at the OMM, and it selectively accumulates on the depolarized/dysfunctional mitochondria and recruits parkin to destroy these mitochondria [66]. Parkin ubiquitinates mitochondria-associated HBx to trigger selective mitophagy. HBx stimulates parkin translocation to mitochondria for degradation of Mfn2 by ubiquitination in a PINK1/parkin-dependent manner in HBV-replicating Huh7 cells transiently transfected with 1.3 mer HBV genome under the control of native promoter [7]. Mfn2 is an outer mitochondrial membrane protein and plays a vital role for mitochondrial fusion. Kim et al. demonstrated that HBV and its encoded HBx protein promoted mitochondrial fragmentation (fission) via Drp1 stimulation, and mitophagy via parkin, PINK1, and LC3B stimulation [7]. They also demonstrated that HBx-induced mitophagy to attenuate mitochondrial apoptosis, suggesting that HBV-induced mitochondrial fission and mitophagy should facilitate cell survival and viral persistence. Huang et al. reported that HBx-induced mitophagy through the PINK1-parkin pathway by increasing mitochondrial LONP1, which plays roles in the unfolded protein response (UPR) in the mitochondrial matrix. This phenomenon has been confirmed in HBV-replicating HepG2.2.2.15 cell line [67,68]. Chi et al. investigated the mitochondrial localization of HBx with its effect on mROS and Δψm, which correlated with their data from HBx transgenic mice and clinical HBV-mediated HCC patients [69]. HBx-induced carcinogenesis in HBx transgenic mice and predominantly localized into mitochondria in stably expressing HepG2 cells and interacted with PINK1 and parkin [69]. Thyroid hormone (TH) simultaneously induces mitochondrial biogenesis and autophagy of HBx-targeted mitochondria through PINK1 induction to suppress HBx-promoted ROS generation and carcinogenesis [69]. They further demonstrated that TH abolished HBx-mediated upregulation of transcription factors, phospho-STAT3, c-Jun, NF-κB, and AP-1 through the increase of mROS and the reduction of Δψm, and these results are consistent with others reports [35,37,69]. These TH/PINK1/Parkin signaling effects on the reduction of HBx- mediated HCC are correlated with clinical cases [69]. HBx promotes stem/progenitor cell markers in HBx-transgenic mice treated with 3,5-diethoxycarbonyl-1,4-dihydrocollidine (DDC) [70]. These results were confirmed by the increase of IL-6/STAT3 and Wnt/β-catenin signaling activities in HBx- transgenic mice [70]. 

It has been reported that HBx interacts with heat shock proteins (HSPs) in mitochondria [71,72]. HSP60 is matured in the mitochondrial matrix after cleavage [73]. Tanaka et al. demonstrated that HBx interacted with endogenous HSP60 in transfected Huh7 cells and enhanced HBx-mediated apoptosis [71]. Although HSP70 is mainly localized at the endoplasmic reticulum, mitochondrial HSP70 plays a vital role for mitochondrial protein folding [74]. HBx binds with the mitochondrial HSP70 and forms a complex with endogenous HSP60 and HSP70 in transfected COS7 cells to fulfill its function [72]. Parvulin 17 (Par17) is targeted to mitochondrial matrix but parvulin 14 (Par14) localizes in the cytoplasm, nucleus, and mitochondria as well [75,76]. Par14 and Par 17, which play roles in protein folding, chromatin remodeling, cell cycle progression and so on, directly interact with HBx and promote HBx translocation to the nucleus and mitochondrial fractions, and upregulate HBV DNA replication [77]. The Par14/17-HBx interaction and colocalization have been confirmed by cell fractionation assay followed by IP and by IFA in transfected HEK293T cells. However, the authors hypothesized that the interaction between Par14/17 and HBx forms a complex with cccDNA (cccDNA-Par14/17-HBx complex) and this complex upregulates the HBV RNA transcription [77]. 

Mitochondrial antiviral signaling protein (MAVS) localizes at mitochondria by its C-terminal transmembrane anchor and mitochondria-associated membranes (MAMs) [78,79]. In addition, MAVS directly interacts with a translocase of the outer mitochondrial membrane 70 (TOM70) during viral infections, and TOM70 associates with HSP90 [80,81]. Several reports have suggested that HBx likely interacts with MAVS and attenuates antiviral immune responses [5,82,83]. Kumar et al. confirmed the interaction of HBx with MAVS in HBx-transgenic mice and transfected HepG2 cells [84]. Wei et al. also demonstrated that HBx interacted with endogenous MAVS in HEK293T cells, which were transfected with Myc-tagged HBx [82]. Parkin is an E3 ligase that plays an important role in protein ubiquitination [85]. HBx stimulates parkin to interact with MAVS through recruitment of the linear ubiquitin assembly complex (LUBAC) for disruption of the MAVS signalosome and for attenuation of IRF3 activation [83]. HBx also bind with parkin in Huh7 cells cotransfected with Flag- HBx and mCherry-Parkin [7]. HBx promotes the degradation of MAVS through ubiquitination and blocks MAVS-mediated IFN-β induction [82]. HBx-mediated MAVS degradation strongly correlated with the results from clinical HCC samples from HBV patients and HBx transgenic mice [82]. Thus virus-induced RIG-I-MAVS signaling is inhibited by HBx, which leads to attenuation of antiviral immune responses of the innate immune system [5,82]. However, a recent study has been conducted by collecting liver tissue samples from chronic HBV patients and demonstrated that HBV does not interfere with the innate immune response [86]. 

Therefore, considering all of the above results, it might be stated that HBx has a role for development of HBV-mediated HCC by modulating mROS, Δψm, and mtDNA through the activation of several transcription factors. As discussed above, several reports also demonstrated that HBx may play a role in mitochondria-mediated cellular apoptosis and attenuation of innate immune response. During these processes, at least a fraction of HBx may localize at mitochondria, and directly or indirectly interacts with many kinds of mitochondrial proteins, and exerts effects on the morphological and functional changes of mitochondria, thereby regulating various kinds of cellular responses which are advantageous to HBV replication (Figure 2). However, a few reports also demonstrated that HBx localized into the nucleus either in Huh7 and HepG2 cells transfected with HBx-expressing plasmids and HBV-infected PHH [87,88]. Kornyeyev et al. showed that HBx was mainly detected in the nucleus in HBV-infected PHH cells. In contrast, the HBx mutant lost the DDB1- binding activity; which was detected in both the cytoplasm and nucleus, suggesting that the nuclear localization of HBx depends on the interaction of HBx with DDB1. Although authors explained that the cytoplasmic localization of HBx may occur as results of saturation of the HBx–DDB1 interaction, because the cytoplasmic HBx was detected only in the highly expressing cells [87].

### 3.2. Polymerase

HBV polymerase (pol) is a multifunctional reverse transcriptase protein, and its ORF overlaps with the three other ORFs, S, X, and C. It has four domains: a terminal protein (TP), spacer, reverse transcriptase (RT), and RNase H domain. Pol plays essential roles in viral εRNA binding to package pgRNA into nucleocapsids, and initiates reverse transcription [89,90]. During HBV replication, pol also interacts with many cellular proteins, including mitochondrial ones [20,91,92,93].

A recent study reported that pol has a mitochondrial targeting signal (MTS) and localizes at the mitochondria in HBV-replicating Huh7 cells transfected with HBV genome under the control of CMV immediate early promoter, but neither core nor pgRNA localizes there during viral DNA replication [20]. Many reports have indicated that pol inhibits innate immune responses by preventing the upregulation of interferon regulatory factor 3 (IRF-3) [92,94]. Liu et al. clearly demonstrated the pol- mediated downregulation of IFN-β through direct interaction with stimulator of interferon genes (STING) in Huh7 cells transfected with plasmid pHBV1.3 or pCMV-HBV [95]. RT domain of pol also drastically reduces the K63-linked polyubiquitination of STING, which is necessary for the activation of IFN induction pathway [95,96]. It has been reported that STING associates with MAMs [5]. Therefore, Unchwaniwala et al. assumed that pol might interact with STING at mitochondria, although they did not discuss the involvement of mitochondria in pol-mediated immune responses in their study (Figure 2) [20]. Again it should be mentioned here, Suslov et al. demonstrated that HBV does not interfere with the innate immune response in chronic HBV-infected patients [86].

### 3.3. HBsAg

The HBV nucleocapsid is surrounded by three surface (envelope) proteins encoded by an ORF preS-S sharing a common S ORF. The large surface protein (LHBsAg, LS) consists of preS1, preS2, and S ORF, and the middle one (MHBsAg, MS) consists of preS2 and S, whereas the smallest one (SHBsAg, SS) contains only S ORF [97]. Subviral particles consisting of MS and SS are produced in greater excess compared to mature viral particles, and all three envelope proteins are required for efficient infectious HBV (Dane particles) formation [11], even though MS is not necessary. HBV attaches to cells by the preS1 regions of the LS, whereas SS is an important determinant for diagnosis [12].

The ubiquitous molecular chaperone GRP78 generally localizes at ER, but a part of GRP78 is also present in mitochondria, especially in the intermembrane space, inner membrane, and matrix under the unfolded protein response (UPR) [98]. ER stress is activated by HBV infection through the ER- associated degradation (ERAD) pathway in persistently replicating HepG2 2.2.15 cells, and reduces the production of envelope proteins [99]. GRP78 is up-regulated in HepAD38 cells, a persistently HBV genome-integrated cell line under the control of a tet operator/CMV promoter and inhibits viral replication through the IFN-β-mediated pathway [100]. However, GRP78 expression was lower in liver tissues of post lamivudine treated clinical patients compared with pretreated cases though it showed cytoplasmic and perinuclear staining pattern [100]. Moreover, Cho et al. demonstrated that GRP78 binds with preS1 of LS in HepG2 cells transfected with a replication competent plasmid pHBV5.2 under the control of autologous regulatory elements, although exact importance of GRP78 in HBV replication has not been clarified [101]. Heat shock protein family A (HSP70) family member 9 (HSPA9), also known as GRP75, is primarily localized in mitochondria but also found at lower levels in the ER [102]. GRP75 is involved in cell proliferation by interacting with p53 [103]. The preS1 region of LS physically binds with the GRP75 in co-transfected COS7 cells, where it is expected to regulate proper folding of HBV envelope proteins [104]. Though, HBV envelope proteins are assembled with naked capsids on the ER-Golgi and/or multivesicular body (MVB) [105,106,107] none of the above studies showed colocalization of HBsAg and GRP75/78 specifically in mitochondria.

SS binds with enoyl-coA hydratase short chain 1 (ECHS1), which is located in the mitochondrial matrix, acts on the fatty acid beta-oxidation pathway, and reduces the ECHS1 expression in hepatoma cells [108]. A study reported that SS interacted with ECHS1 in the cytoplasm of co-transfected 293FT cells, and that co-existence of ECHS1 and SS induced apoptosis by decreasing Δψm and upregulating pro-apoptotic proteins (Bad, Bid, Bim, etc.) in HepG2 cells stably expressing HBs [109]. Jumping translocation breakpoint (JTB) protein is overexpressed in HCC, and is thought to play an important role in oncogenesis in the liver [110,111]. SS is reported to bind with JTB in transfected HepG2 cells and reduce the mitochondrial localization of JTB, and to inhibit phosphorylation of p65, a subunit of the NF-κB complex, implying that SS might have a role in HCC progression (Figure 2) [110]. 

These reports suggest that some SS is localized in the mitochondria, where it associates with a few mitochondrial proteins and affects their function.

### 3.4. Core

HBV core (HBc) is a small protein, consisting of 183 amino acids produced by ORF C, which is self-assembled to form viral capsids [112,113,114]. The hTid1, a family protein of HSP40, predominantly localizes in mitochondria and is involved in apoptosis [115,116,117]. The hTid1 interacts with HBc and reduces viral replication by degradation of HBc and HBx proteins (Figure 2) [118]. The authors confirmed the hTid1-HBc interaction in Huh7 cells co-transfected with pcDNA6/V5-HisA (hTid1) and pHBcHA (HBc) and investigated these effects in HepG2 cells transfected with replication competent plasmid pHBV1.3 under the control of CMV promoter [118].

## 4. Conclusions and Perspectives

According to most of the studies discussed above, at least some fraction of HBx may localize and/or associate with the mitochondria and associated proteins to disturb mitochondrial dynamics/signaling and could play a momentous role in HBV-mediated HCC formation. Many host proteins/factors such as transcriptional, anti-apoptotic, pro-apoptotic, and innate immune-related proteins are thought to be involved during HBx-mitochondria mediated pathogenesis. There is a possibility that another functional protein, pol, could benefit from mitochondria by suppressing STING-mediated antiviral signaling. As for HBsAg, it is ubiquitous in and around mitochondria and could interact with some factors therein. However, most of these studies regarding HBV and mitochondria have been conducted in an overexpression system, and a few reports also showed that HBx predominantly localized into nucleus even in infection condition. Therefore, such phenomena/mechanisms should be verified in infection systems and also in HBV-infected patients. 

In conclusion, HBV has been suggested to exert substantial effects on mitochondria to change mitochondrial dynamics/signaling, leading to HCC development. Further investigations of the relationships between HBV and mitochondria are needed to improve our understanding of mitochondrial involvement in the HBV life cycle. Such research could open a door to novel therapeutic strategies directed at mitochondria.

## Figures and Tables

**Figure 1 viruses-12-00175-f001:**
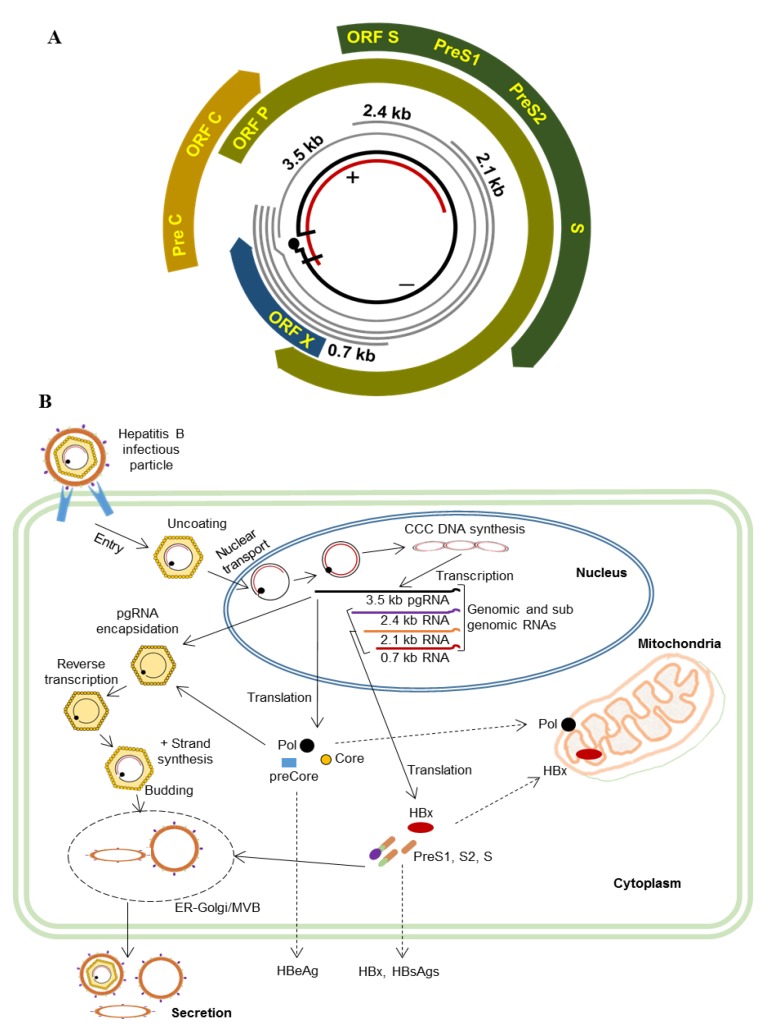
Hepatitis B virus (HBV) genomic map and an overview of the HBV life cycle. (**A**). HBV genomic map. The partially double-stranded DNA encodes four overlapping open reading frames (ORFs), preC-C, P, preS-S, and X. The ORF P overlaps the other three open reading frames (ORFs). (**B**). An overview of the HBV life cycle. HBV infects hepatocytes through preS1-NTCP interaction followed by uncoating and is transported to the nucleus where cccDNA is formed. The cccDNA acts as a template for transcription of the 3.5 kb pregenomic RNA (pgRNA), and the 2.4 kb, 2.1 kb, and 0.7 kb subgenomic RNAs. The pol translated from pgRNA is encapsidated along with pgRNA, then reverse-transcribed, and the partially double-stranded DNA genome is formed. The core particle is enveloped in ER-Golgi/MVB and secreted into the extracellular space.

**Figure 2 viruses-12-00175-f002:**
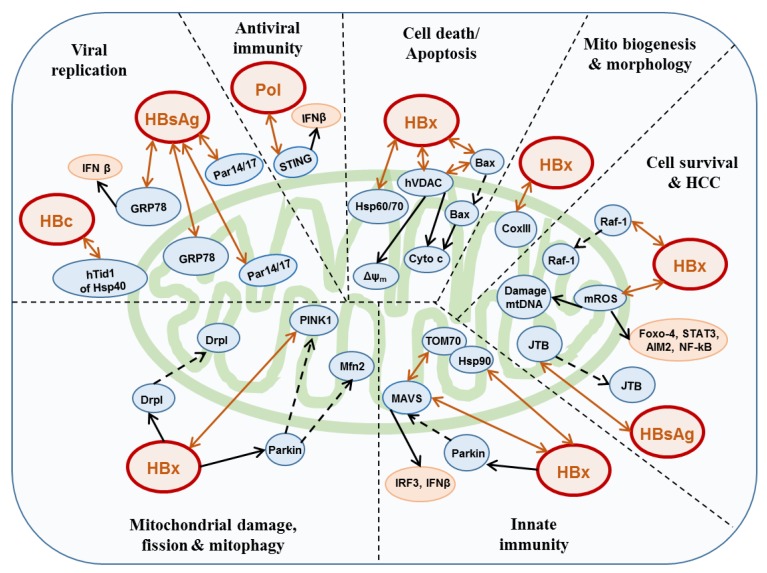
Interactions between HBV proteins and mitochondrial proteins. HBx interacts with different mitochondrial proteins and/or translocates several cytosolic proteins into mitochondria, which affects mitochondrial fission, morphology, and biogenesis and leads to mitophagy/dysfunction. Innate immunity is lost due to decreased activities of IRF3 and IFN-β through MAVS/parkin/HBx interaction. Cell death/apoptosis may occur due to loss of Δψm and release of cytochrome c induced by HBx. Increase in mROS due to HBx causes mtDNA destruction and activation of oncogenic transcription factors, which might lead to HCC development. Pol interacts with STING and reduces the antiviral immunity through suppression of IFN-β. HBsAg interacts with and degrades GRP78 and up-regulates IFN-β to suppress HBV replication. Viral replication is also suppressed by degradation of HBc and HBx through interaction between HBc and hTid1. HBx translocates Raf1 into mitochondria. HBsAg binds with JTB and reduces the mitochondrial localization of JTB. Double arrows indicate interactions and dashed arrows indicate translocations.

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
