# Peer review of "Impact of the Interaction of Hepatitis B Virus with Mitochondria and Associated Proteins"

_viruses, 2020, doi:10.3390/v12020175_

Round 1
Reviewer 1 Report
The authors did revise the manuscript significantly by expanding scientific content and improving English. Not the text is quite clear and readable.
In particular, they tried to report which studies were obtained in which system. But this is still not clear for localization of HBV proteins. But since they can have different localization upon overexpression and during infection, this requires further clarification.
A passage on lines 190-210: Grp75 is not just ER- on mitochondria-residing protein but a protein that links both organelles in MAMs. So, the P protein could affect MAM integrity and thus have an impact on mitochondrial functions. This could be worth mentioning here
Line 84: Please indicate what is meant by the phrase „recent papers”
Please revise the phrase on lines 170-171.
Author Response
Thank you very much for reviewing our manuscript. We have revised the manuscript according to your suggestions and comments.
Point 1: In particular, they tried to report which studies were obtained in which system. But this is still not clear for localization of HBV proteins. But since they can have different localization upon overexpression and during infection, this requires further clarification.
Response 1: We have added the description of systems in detail throughout the current version of the manuscript. Please check the purple-colored text in the revised/latest version of the manuscript. Furthermore, we also correlated the points by discussing the data from clinically infected patients with their limitations as well.
Point 2: A passage on lines 190-210: Grp75 is not just ER- on mitochondria-residing protein but a protein that links both organelles in MAMs. So, the P protein could affect MAM integrity and thus have an impact on mitochondrial functions. This could be worth mentioning here.
Response 2: This is a very important point. Polymerase protein could affect MAM integrity and thus have an impact on mitochondrial functions which have been discussed in the Polymerase section (3.2). We have not found any published report on interaction/relation between grp75 and pol so far we searched. Therefore, we omitted the discussion regarding grp75 and pol.
Point 3: Line 84: Please indicate what is meant by the phrase „recent papers”.
Response 3: We have rewritten the sentence (Lines 85-86).
Point 4: Please revise the phrase on lines 170-171.
Response 4: We corrected the phrase including the paragraph of the manuscript (Lines 229-243).
Reviewer 2 Report
The review can be accepted in the present form
Author Response
Thank you very much for accepting the revisions.
Reviewer 3 Report
The authors have made substantial efforts towards editing the manuscript and addressing the concerns of the reviewers. The manuscript has significantly improved, making for a much more comprehensive and organized review article. The updated manuscript is suitable for publication in Viruses.
Author Response
Thank you very much for your appreciation and for accepting the revisions.
This manuscript is a resubmission of an earlier submission. The following is a list of the peer review reports and author responses from that submission.
Round 1
Reviewer 1 Report
The Review article entitled “Hepatitis B viral proteins and mitochondrial dynamics” written by Md. Golzar Hossain covers the literature surrounding how Mitochondria or mitochondrial proteins are alerted by Hepatitis B virus infection. The effects of virus infection on mitochondrial dynamics and activity is an important topic as mitochondria are central in regulating many cellular pathways. Moreover, this topic has not been reviewed giving an article written on this topic the potential to be high relevance to the field. However, the current version of this manuscript requires an extensive amount of work before it would be suitable for publication.
Major concerns
The most pressing concern with the manuscript is that is has a large number of spelling and grammatical errors that make many ideas or topics difficult to understand or interpret. The author should consider having the paper copy edited. Additionally, there are a number of mistakes in the descriptions of mitochondria and mitochondrial processes as well as protein localization. The author should be clear on the literature surrounding mitochondrial function so as to not make misleading in incorrect statements. Additionally, the focus of the manuscript should be shifted more towards how the virus affects mitochondrial dynamics and function rather than localization of viral proteins to the mitochondria (see specific points for details). The author should also refrain from making broad conclusions that are not supported by the evidence in the literature they are reviewing.
Specific points
The first sentence in the introduction (line 27) starts by stating that “Mitochondria are self-replicating double membrane organelles”. Though the Mito have their own genome, they require many factors coded from the nuclear DNA in order to grow and divide. Therefore, mitochondria should not be described as “self-replicating”. In line 63, the author states that “more or less all HB viral proteins localize into mitochondria”. This is a confusing statement especially since the papers cited do not show localization of HBV proteins to the mitochondria. In fact Kim et al. show that HBsAg is not localized to the mitochondria. Though there is some evidence that a subpopulation of HBx and HBV polymerase might localize to the mitochondria, there is little evidence that core or HBsAgs reside there. In the sentence starting on line 120, the author states that HBx-mediated down regulation of MAVS “confirms the localization of HBx onto the OMM”. This does not confirm localization to the OMM. This is a phenotypic and physical connection with MAVS which says nothing about HBx protein localization or where this interaction occurs. In line 183 the author claims that the viral polymerase might go to the mitochondrial matrix for proper folding. Knowing what we do about protein folding and protein transport into the mitochondrial matrix, this is not a feasible model. There is little evidence for protein export from the Mitochondria. The author should provide more evidence for this type of conclusions before making them in a review article. In line 204 the author states that HBsAgs might concentrate around the mitochondria (MAMs or OMMs); however, the data provided by Kim et al. (Ref #6) do not see any HBsAg accumulation in mitochondrial fractions. Additionally, none of the references provided in the following paragraph (lines 205-213, ref #60-63) show HBsAg at the mitochondrial. Rather they suggest HBsAg is responsible for directing several factors away from the mitochondria. The author needs to address this discrepancy.
Reviewer 2 Report
The review aims at describing the effects of the HBV proteins on mitochondria.
A review on a topic like this should include a more systematic review of the literature available, preferably it should be described how the author gathered this literature by a systematic search approach
The title does not cover the contents, the review does not focus on the effect of HBV proteins on mitochondrial dynamics, but on the effects of HBV proteins interacting with the mitochondria and associated proteins.
The abstract suggests that mitochondria can be targeted to reduce HBV replication and/or pathology, but this is not supported by the review. only one study is mentioned (ref. 30), this is insufficient.
The review seems to 'push' the notion that HBx localises to the outer mitochondrion membrane, but this notion is insufficiently supported by the referenced literature.
The canonical function of HBx, to drive HBV RNA transcription, is not introduced. Interactions witht eh mitochondria, and the relation to HBV replication, should be considered in this perspective.
The interactions between HSPs and HBV proteins do not involve mitochondria or mitochondria function (!). HSPs are NOT mitochondrial chaperones, as stated by the author (line 162, 177). The review critically depends on this 'notion'.
Some sentences cannot be interpreted (e.g. 111-112), and if the author wishes to make such statements they should be supported by literature.
lines 121-122: this statement is false, one cannot conclude this from the data referenced.
All in all, it is speculative if the effects of HBx on mitochondria depend on interactions with specific proteins on the mitochondria, as the author suggests, or stem from another interaction that results in the modulations described.
Reviewer 3 Report
The main topic of this review is of great interest since a review summarizing the direct impact of HBV proteins on mithocondria functionality and signaling pathways is still missing. This issue is particularly relevant since several papers suggest that altered mithocondria dynamics can be involved in mechanisms underlying HCC onset. However, the review is difficult to follow particularly for the pletora of English mistakes, disseminated throughout the whole text. Some sentences are meaningless. The different data from the already published studies are reported in the text without a precise order and in absence of a valid interpretation. For all the aspects I retain that the review is not suitable for publication in Viruses.
Reviewer 4 Report
The manuscript by Hossain aims to summarize current data on interaction of hepatitis B virus (HBV) and its proteins with mitochondria and its consequences. The topic is very interesting indeed. However, the manuscript has a number of drawbacks that make it not acceptable in its current form.
First of all, English is quite poor. The whole manuscript has to be edited by a native-speaker. In the current version, numerous flaws take all attention of the reader from scientific ideas. I hate to comment the text of manuscripts but here numerous phrases have to be rewritten.. Most of research has been carried out in cells overexpression individual virus proteins. So, it should be acknowledged that most of the data discussed in the review still have to be validated in infectious models. And the author needs to carefully show which data were obtained in which system. The text makes the reader think that HBV proteins are localized solely in mitochondria. However, this is not true. For example, there are data showing that HBx can be found in the ER and even nucleus, and HBe as well is an ER-residing/associated protein. So again, protein localization should be discussed more carefully. The title of the review declares that it should discuss mitochondria dynamics. But there are only scarce words about fusion/fission and mitophagy. The author should consider expanding the topic by discussing what is known about association of fusion/fission with metabolism and respiration in general, how dynamics is affected by interaction of mitochondria with lipid droplets etc. And probably how this knowledge outside virology field may impact research for HBV. Poor quality (resolution) of the Figure 1, especially of its panel A. Human VDAC should be abbreviated as (hVDAC).